# A stretchable, mechanically robust polymer exhibiting shape-memory-assisted self-healing and clustering-triggered emission

Xiaoyue Wang [1,2], Jing Xu[1,2], Yaoming Zhang [1], Tingmei Wang[1,2], Qihua Wang[1,2,3], Song Li [1] ✉, Zenghui Yang [1] ✉ & Xinrui Zhang [1] ✉

Self-healing and recyclable polymer materials are being developed through extensive investigations on noncovalent bond interactions. However, they typically exhibit inferior mechanical properties. Therefore, the present study is aimed at synthesizing a polyurethane–urea elastomer with excellent mechanical properties and shape-memory-assisted self-healing behavior. In particular, the introduction of coordination and hydrogen bonds into elastomer leads to the optimal elastomer exhibiting good mechanical properties (strength, 76.37 MPa; elongation at break, 839.10%; toughness, 308.63 MJ m$^{-3}$) owing to the phased energy dissipation mechanism involving various supramolecular interactions. The elastomer also demonstrates shape-memory properties, whereby the shape recovery force that brings damaged surfaces closer and facilitates self-healing. Surprisingly, all specimens exhibit clustering-triggered emission, with cyan fluorescence is observed under ultraviolet light. The strategy reported herein for developing multifunctional materials with good mechanical properties can be leveraged to yield stimulus-responsive polymers and smart seals.

Polyurethanes (PUs) are extensively used in scientific research and engineering applications owing to their decent chemical stability, wear resistance, heat insulation, and noteworthy molecular designability[1-5]. However, conventional PUs are increasingly failing at satisfying human needs; therefore, ingenious molecular designs are being implemented to obtain multifunctional materials[6-13]. For instance, some PU block copolymers with diverse functions such as self-healing, shape-memory, and recycling characteristics have recently been developed through skillful molecular design[14-21]. Self-healing can be achieved through dynamic reversible interactions that are induced by external stimuli, leading to molecular chain exchange and topological rearrangements. For example, Liu et al.[22] synthesized a PU containing side-chain hydrogen bonds using a T-shaped chain extender. Owing to the flexibility of the side-chain hydrogen bonds, scratches healed after 6 h

under ambient conditions. Sun et al. [23]. Synthesized a high-mechanical-performance polyurethane–urea (PUU) elastomer with hydrogen-bond arrays, whose cracks healed upon exposure to 90 °C for 12 h. Zhang et al.[24] prepared an intrinsically black poly-(urea-oxime urethane) thermoset that could be triggered by a near-infrared laser, with dynamic exchange of the oxime-carbamate bonds helping efficiently weld two films. Moreover, dynamic reversible interactions have also been used as reversible switches or net points to realize the shape-memory effect[6,8,25,26]. Anthamatten et al.[27] were the first to demonstrate that the thermal reversibility of self-complementary hydrogen bonds within a material endows it with shape-memory properties. Schubert et al.[28] developed a supramolecular shape-memory polymer by introducing two metal coordination bonds into a material, in which a strong terpyridine complex and a labile histidine complex acted as a

[1]Key Laboratory of Science and Technology on Wear and Protection of Materials, Lanzhou Institute of Chemical Physics, Chinese Academy of Sciences, Lanzhou 730000, China. [2]Center of Materials Science and Optoelectronics Engineering, University of Chinese Academy of Sciences, Beijing 100049, China. [3]State Key Laboratory of Solid Lubrication, Lanzhou Institute of Chemical Physics, Chinese Academy of Sciences, Lanzhou 730000, China. ✉ e-mail: lisong@licp.cas.cn; yangzh@licp.cas.cn; xruiz@licp.cas.cn

stable phase and switch, respectively. However, the aforementioned materials typically exhibit inferior mechanical properties. Consequently, supramolecular interactions are being incorporated into materials through molecular design. Energy dissipation mechanisms originating from noncovalent bond interactions contribute to improving the mechanical properties of PUs. Xin et al.[29] fabricated poly(boron−urethanes) with good mechanical performance by enabling the formation of multiple hydrogen bonds. Ding et al.[30] developed a supramolecular PU elastomer by incorporating a dynamic covalent boronic ester, hydrogen bonds, and boron−nitrogen coordination, with the latter two dynamic reversible interactions being sacrificial bonds that provided high toughness and rebound resilience. Liu et al.[31] introduced disulfide bonds and hydrogen-bonded urea groups into PU, resulting in high toughness and a high fracture energy. However, the development of multifunctional polymers exhibiting notable shape-memory, self-healing, and mechanical attributes remains challenging.

In this study, a supramolecular PUU with excellent mechanical properties was developed by utilizing abundant coordination and hydrogen bonds as sacrificial bonds for phased energy dissipation. The synthesized elastomers (denoted as PUU-X, where X is the $Zn^{2+}$/pyridine molar ratio) exhibited shape-memory properties that could be fixed at −50 °C and recovered to the original shape at 50 °C. Furthermore, the shape-recovery effect drove cracks to close gradually; the exchange and recombination of nondynamic covalent bonds at the fracture surface were promoted, leading to cracks progressing from active closure to healing. Intriguingly, the material also exhibited clustering-triggered emission (CTE)−specifically, cyan fluorescence−upon being irradiated with ultraviolet (UV) light. Notably, this phenomenon corroborated the existence of hydrogen bonds, as they prompted the luminescent groups to cluster and fluoresce.

## Results

Synthesis and characterization of PUU-X. The prepolymer method was adopted to prepare PUU-X (Fig. 1a). Polycarbonate diol (PCDL; −OH) was reacted with hexamethylene diisocyanate (HDI; −NCO) for 3 h in a $N_2$ atmosphere at 60 °C. The resulting product was then reacted with a large amount of succinic dihydrazide (SDH; −NH$_2$) solution for 2 h, yielding −NH$_2$-terminated molecular chains. PUU-0 was subsequently obtained after the resulting product was reacted with 2,2'-bipyridine-5,5'-dicarboxaldehyde (BIDI; −CHO). The PUU-0 solution was then mixed with zinc trifluoromethylsulfonate (Zn(OTf)$_2$)/pyridine solutions with different molar ratios, yielding specimens with shape-memory and self-healing properties (denoted as PUU-X). The synthesis of the PUU-X samples was verified by solid-state $^{13}C$ nuclear magnetic resonance (NMR) analysis (see Supplementary Figs. 2−6) and Fourier-transform infrared (FTIR) spectroscopy (Fig. 1b). The peaks at 1738 and 1242 cm$^{-1}$ were associated with the stretching vibrations of C = O in the ester carbonyl group and C−O in the ester group, respectively. The peak corresponding to C = O in urea appeared at 1690 cm$^{-1}$, whereas that of the acyl hydrazone bond (C = N) appeared at 1662 cm$^{-1}$; moreover, the peak at 1585 cm$^{-1}$−which attributed to the pyridine group of PUU-0−disappeared gradually with increasing $Zn^{2+}$ content, indicating the formation of $Zn^{2+}$−pyridine coordination bonds. The thermal stability of the PUU-X specimens was characterized by thermogravimetric analysis (Fig. 1c and Table 1; $T_{5\%}$: temperature corresponding to 5% mass loss of PUU-X). The polymer decomposed almost completely at ~600 °C; however, the residual mass of the PUU increased gradually to 2.33%, 3.54%, 5.29%, 6.26%, and 7.31%, respectively, upon the incorporation of $Zn^{2+}$ (that is, Zn (OTf)$_2$). However, the thermal stability of PUU-X was not significantly affected by the addition of $Zn^{2+}$, although it did modify other properties to some extent due to the physical crosslinks derived from coordination bonds. X-ray

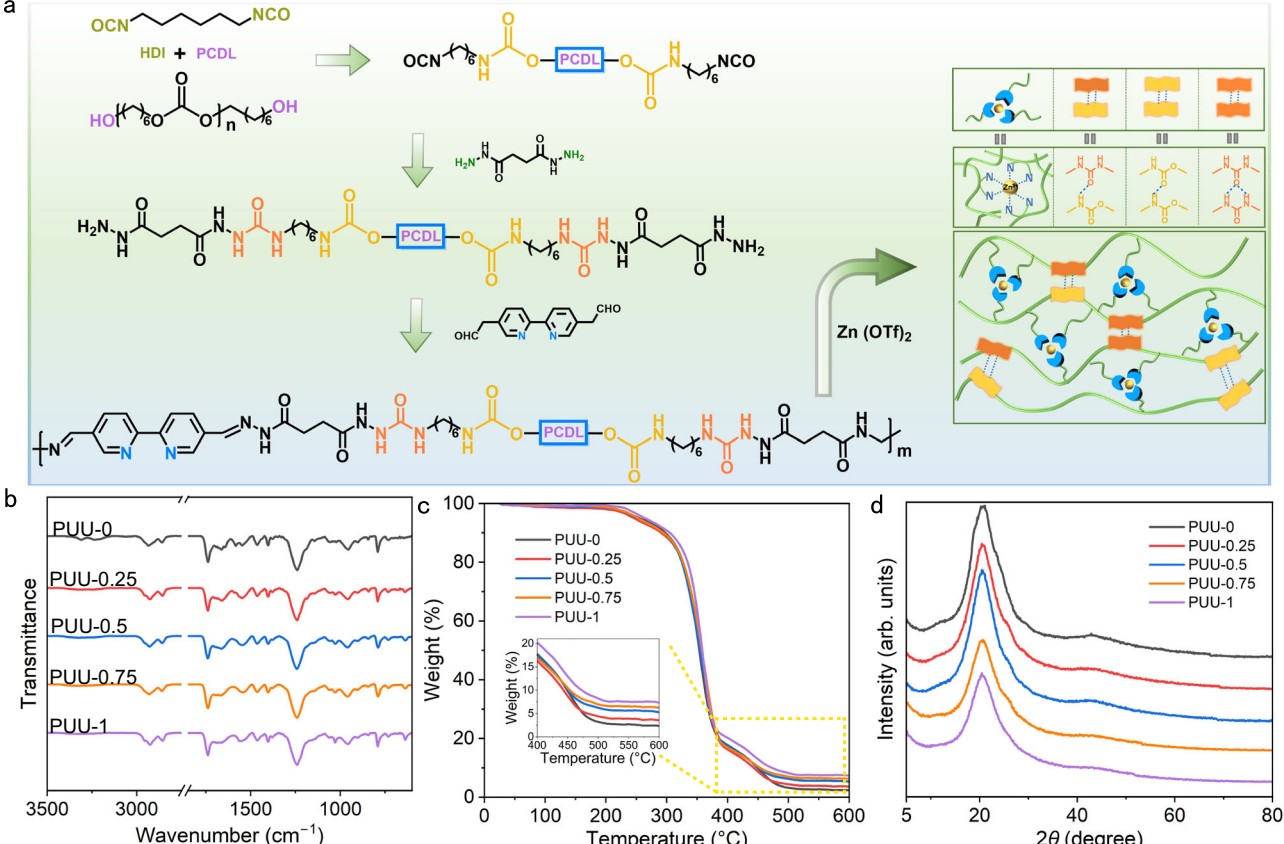

**Fig. 1 | Synthesis route and characterization of PUU-X elastomers. a** Schematic illustrating the synthesis of PUU-X elastomers. **b** FTIR spectra, **c** thermogravimetry curves, and **d** XRD patterns of PUU-X.

**Table 1 | Physical and chemical data of the synthesized PUU-X samples**

|          | $T_{5\%}$ (°C) | $T_g$ (°C) | Tensile strength (MPa) | Strain at break (%) | Toughness (MJ m$^{-3}$) | $R_f$ (%) | $R_r$ (%) |
|----------|------------|--------|------------------------|---------------------|-------------------------|-----------|-----------|
| PUU-0    | 255.07     | −36.08 | 39.74 ± 6.78           | 736.708 ± 33.11     | 182.53 ± 26.95          | 87.19%    | 97.19%    |
| PUU-0.25 | 246.61     | −36.25 | 51.23 ± 1.05           | 683.84 ± 35.82      | 185.98 ± 9.33           | 82.18%    | 96.20%    |
| PUU-0.5  | 252.21     | −36.28 | 76.52 ± 2.28           | 801.44 ± 52.80      | 296.90 ± 25.03          | 88.64%    | 97.31%    |
| PUU-0.75 | 253.01     | −36.22 | 47.34 ± 0.91           | 562.96 ± 1.81       | 139.17 ± 0.01           | 87.64%    | 97.18%    |
| PUU-1    | 264.49     | −36.66 | 42.67 ± 2.41           | 614.03 ± 17.48      | 136.5 ± 5.04            | 88.15%    | 94.24%    |

The values of tensile strength, strain at break and toughness are the mean ± standard deviation of $n = 3$ independent experimental repeats.

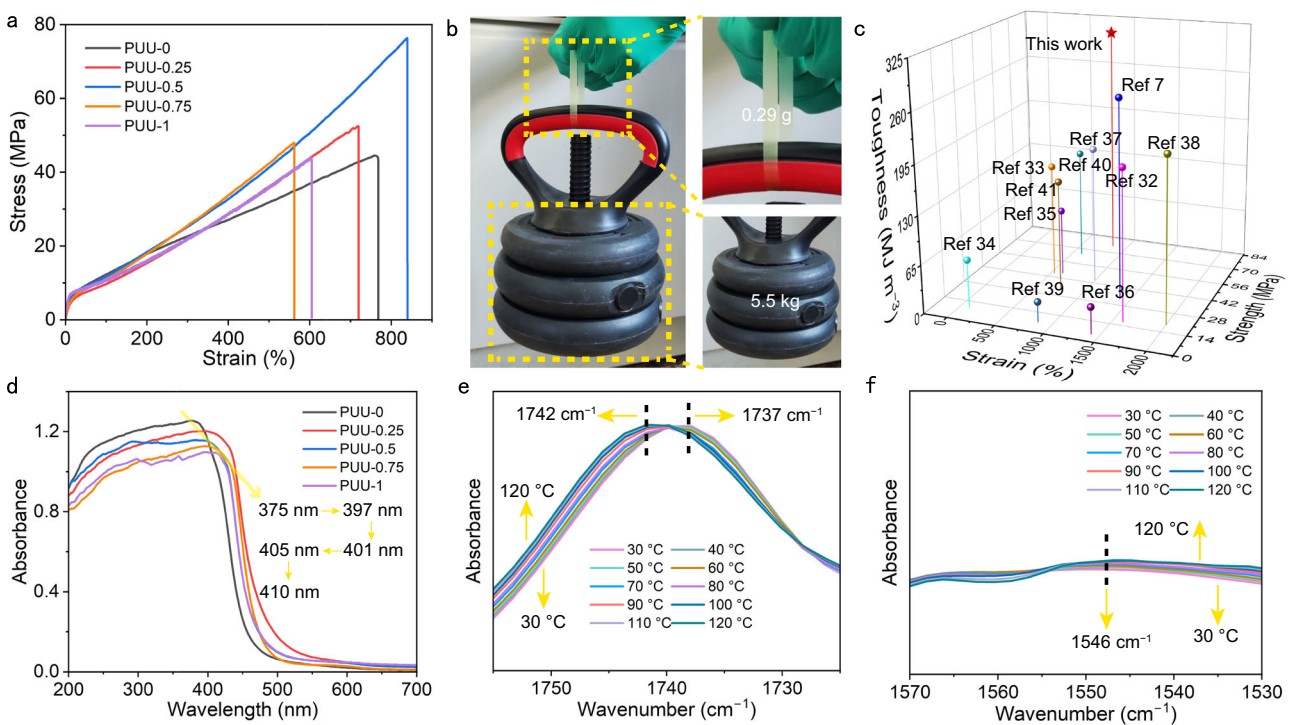

**Fig. 2 | Mechanical properties, UV−vis spectrophotometry, and in situ variable-temperature FTIR spectroscopy of the PUU-X specimens. a** Stress–strain curves of PUU-X. **b** Photographs of PUU-0.5 (0.29 g) sustaining a load of 5.5 kg. **c** Comparison between PUU-0.5 and similar previously reported specimens in terms of strength, strain, and toughness. **d** UV−vis absorption spectra of the PUU-X samples. **e, f** The in-situ temperature-dependent FTIR of PUU-0.5.

diffractometry (XRD) was performed to monitor the crystallization behavior of PUU-X (Fig. 1d). The broad peaks at 20° demonstrated that none of the PUU-X samples had crystalline structures. This result indicates that the shape-memory properties of PUU-X were independent of the crystalline melting temperature.

Mechanical properties of PUU-X. The enhancement of mechanical properties in PUU-X by hydrogen and coordination bonds was confirmed through tensile tests (Fig. 2a, Supplementary Fig. 7, and Table 1). $Zn^{2+}$-free PUU-0 exhibited the most inferior tensile strength (44.53 MPa), elongation at break (760.12%), and toughness (201.58 MJ m$^{-3}$) among the specimens owing to its linear structure. Nevertheless, the mechanical properties were improved upon the addition of $Zn^{2+}$, with the stress, elongation at break, and toughness exhibiting increasing trends initially (Supplementary Fig. 8). In particular, PUU-0.5 exhibited a 1.71-fold higher stress (76.36 MPa), 1.1-fold higher elongation at break (839.10%), and 1.53-fold higher toughness (308.63 MJ m$^{-3}$) than those of PUU-0; more importantly, it exhibited enhanced strength without sacrificing stretchability. Moreover, PUU-0.5 (0.29 g) could lift 5.5 kg of weight (18965 times the sample weight; Fig. 2b) and exhibited good comprehensive mechanical properties compared with those reported previously[7,32–41] (Fig. 2c). The mechanical performance of PUU-X was noticeably enhanced upon the addition of $Zn^{2+}$,

suggesting that the crosslinking derived from the coordination bonds could reform them, based on the physical crosslinking density increasing with escalating $Zn^{2+}$ content. However, the mechanical properties of PUU-X deteriorated upon further addition of $Zn^{2+}$, presumably owing to the agglomeration of the physical crosslinks and the consequent stress concentration.

UV−vis spectrophotometry and in situ variable-temperature FTIR spectroscopy were performed to corroborate the existence of intermolecular interactions in the synthesized materials[42,43]. The absorption peaks of the $Zn^{2+}$-containing materials were red-shifted compared with those of PUU-0 (Fig. 2d) owing to the $Zn^{2+}$−pyridine coordination. The maximum absorption peak red-shifted progressively with increasing $Zn^{2+}$ content (375 nm→397 nm→401 nm→405 nm→410 nm). Different degrees of $Zn^{2+}$−pyridine complexation were achieved with increasing $Zn^{2+}$ content, which enhanced the electron ionization of the pyridine-ring-formed π−π conjugate system and reduced the energy required for transitioning, leading to a gradually red shift[43]. Noteworthy changes were also observed in the acquired FTIR spectra (see Supplementary Fig. 9, Fig. 2e, f). For instance, the C = O peak shifted from 1737 to 1742 cm$^{-1}$ and enhanced in intensity with increasing temperature (Fig. 2e). Furthermore, the intensity of the peak at 1546 cm$^{-1}$ (corresponding to C-N) enhanced with rising temperature (Fig. 2f). These

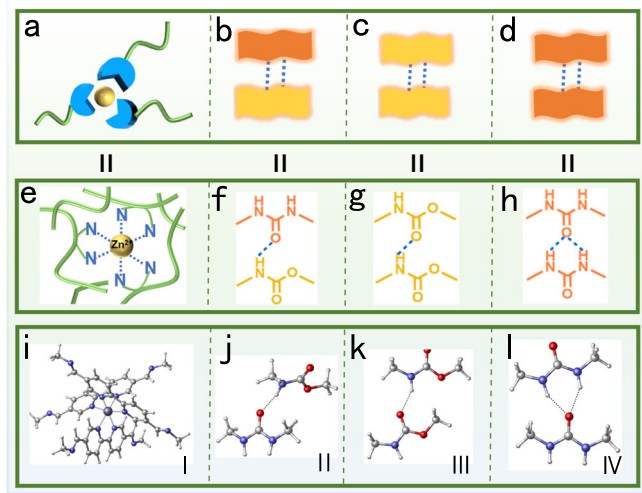

**Fig. 3 | Abridged general view, molecular structure, and optimized configuration of the various noncovalent bond. a, e, i** Abridged general view, molecular structure, and optimized configuration of the coordination bonds. **b–d** Abridged general view, **f–h** molecular structures, and **j–l** optimized configurations of different hydrogen bonds.

### Table 2 | Bonding energies of the coordination and hydrogen bonds

| Type | I | II | III | IV |
|---|---|---|---|---|
| Bonding energy | 49.80 kcal mol⁻¹ | 8.60 kcal mol⁻¹ | 7.97 kcal mol⁻¹ | 11.09 kcal mol⁻¹ |

carbamate–carbamate interactions (7.97 kcal mol⁻¹); these results are consistent with the experimentally determined mechanical properties. The coordination bond dissociation enabled the material to absorb more energy during tensile stretching, certifying that PUU-0.5 exhibited good mechanical properties from a macroscopic perspective.

To elucidate the mechanism governing the evolution of noncovalent interactions during stretching, the materials were strained to different degrees and subjected to small-angle X-ray scattering (SAXS) analysis. One-dimensional (1D) intensity profiles and 2D patterns were acquired (Fig. 4a, b, respectively). Owing to the existence of coordination and hydrogen bonds, the material exhibited a microphase-separated structure in the initial state and a scattering peak at $q = 0.48$. With increasing strain, the hydrogen and coordination bonds gradually dissociated, and the microphase-separated structure was destroyed. This was evidenced by the disappearance of the diffraction peak, a decrease in the diffraction intensity, and a gradual change in the isotropic scattering halo pattern from round to oval to diamond. When the material was heated at 50 °C to restore the original state, the microphase-separated structure was re-formed and the diffraction peak reappeared owing to the re-association of the coordination and hydrogen bonds.

Combining the results of the theoretical calculations and SAXS analysis, a mechanism underlying the evolution of the material during tensile elongation was proposed (Fig. 4c). Due to the presence of coordination and hydrogen bonds, the material initially appears as a microphase-separated structure (Fig. 4c-I and II), and the hydrogen bonds with lower bonding energy dissociate first (Fig. 4c-III). With a further increase in force, the hydrogen bonds with higher bonding energy rupture, followed by the eventual fracture of coordination bonds (Fig. 4c-IV).

**Thermo-mechanical and shape-memory attributes of PUU-X.** Dynamic mechanical analysis (DMA) was performed to determine the glass transition temperature ($T_g$), storage modulus ($E'$; see Supplementary Fig. 12), and loss factor (tan $\delta$; Fig. 5a) of the PUU-X specimens. The decline of $E'$ commenced at around −70 °C and showed a drastic decrease from 1870 to 550 MPa near −36 °C. Moreover, the $T_g$ of all the PUU-X specimens was estimated to be ~ −36 °C. Based on these results, an appropriate temperature range was selected for the shape-memory tests, which were conducted by DMA for all the PUU-X samples. For instance, PUU-0.5 was equilibrated at 50 °C and then subjected to a 2 N tensile load and maintaining the temperature for 5 min (Fig. 5b). The temperature was then cooled to below $T_g$ for 30 min, and the sample was unloaded to obtain the temporary shape because the molecular chains were frozen. The shape fixation rate ($R_f$) was calculated using Eq. 1 as 88.64%. Owing to the heat-induced molecular chain movement, the coordination and hydrogen bonds recombined, and the sample returned to the original shape upon being heated to 50 °C, with a shape recovery rate ($R_r$) of 97.31% (Eq. 2). The shape-memory curves, $R_f$ data, and $R_r$ values of the other materials are presented in Supplementary Figs. 13–16 and Table 1. To illustrate the shape-memory effect of PUU-0.5 (Fig. 5c), the sample was cut into a hand shape and heated to 50 °C. Subsequently, a few fingers of the hand-shaped sample were bent, and this configuration was fixed at −50 °C. The fingers recuperated slowly at 50 °C over ~20 s, confirming the tendency of PUU-0.5 to exhibit good shape-memory properties. Additionally, the original shape of PUU-0.5 could be

trends were presumably due to the gradual dissociation of the coordination bonds when the material was heated and the increase in the number of pyridine groups released. Two-dimensional infrared correlation spectra (2D-COS) of PUU-0.5 (see Supplementary Figs. 10 and 11) were acquired to validate these results. The synchronous spectrum of the sample (see Supplementary Fig. 10) displayed four main autopeaks−(1747, 1747), (1734, 1734), (1712, 1712), and (1653, 1653)−whereas the asynchronous spectrum (Supplementary Fig. 11) showed five main cross-peaks−(1747, 1734), (1747, 1653), (1734, 1712), (1734, 1653), and (1712, 1653). According to Noda's rule[44,45], the temperature sensitivity of each peak during the heating, ordered from fast to slow, is 1747 cm⁻¹ > 1734 cm⁻¹ > 1712 cm⁻¹ > 1653 cm⁻¹. The 1653 and 1712 cm⁻¹ peaks mainly correspond to the C = N affected by coordination bonds and the ordered C = O influenced by hydrogen bonds, respectively. Moreover, the 1734 and 1747 cm⁻¹ peaks are linked to the semi-ordered C = O partially affected by hydrogen bonding and the free C = O without hydrogen bonding, respectively. Considering the original infrared spectral data acquired in the 1600–1800 cm⁻¹ band, the relative intensities of the absorption peaks at -1710, 1747, and 1653 cm⁻¹ increased steadily with increasing temperature, indicating that the hydrogen and coordination bonds in PUU-X were continuously ruptured, thereby forming freer C = O, N–H, and C = N bonds. Overall, these results collectively validate the existence of hydrogen and coordination bonds in the material.

The molecular chain motion during stretching and the diverse contributions of various nondynamic covalent bonds to the mechanical properties of PUU-0.5 were further assessed by performing quantum chemical calculations and estimating the bonding energies of several hydrogen bonds and metal coordination links (Fig. 3). Figure 3a–h depict schematics and molecular structures of several hydrogen bonds and metal coordination links, focusing on three types of hydrogen-bond-forming interactions: urea–carbamate, carbamate–carbamate, and urea–urea interactions (Fig. 3f–h, respectively). Optimized structures of the aforementioned noncovalent bonds were acquired (Fig. 3i–l; see Supplementary Information for Supplementary computational details). The calculated data (Table 2) revealed that the bonding energy for the various noncovalent bond interactions in the material was ordered as follows: coordination bond (49.80 kcal mol⁻¹) > urea–urea interactions (11.09 kcal mol⁻¹) > urea–carbamate interactions (8.60 kcal mol⁻¹) >

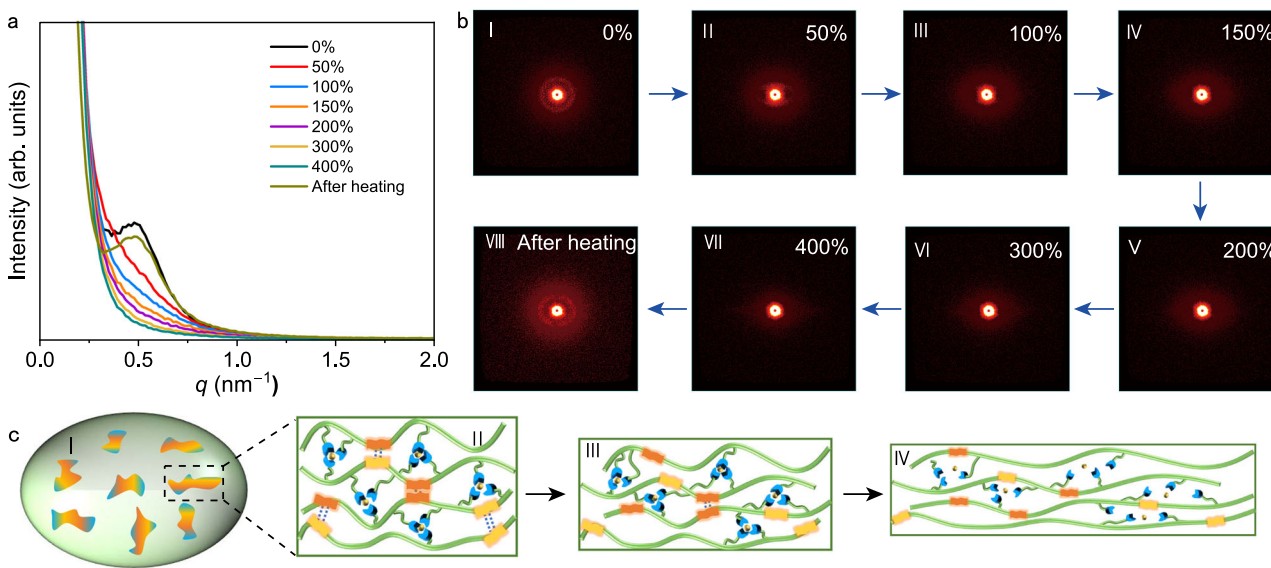

**Fig. 4 | SAXS analysis and mechanism of PUU-0.5 behaving under tensile strain. a** 1D SAXS profiles and **b** 2D SAXS patterns of PUU-0.5 subjected to different stretching strains. **c** Illustration of the molecular chain motion and nondynamic covalent bond dissociation in PUU-0.5 during stretching.

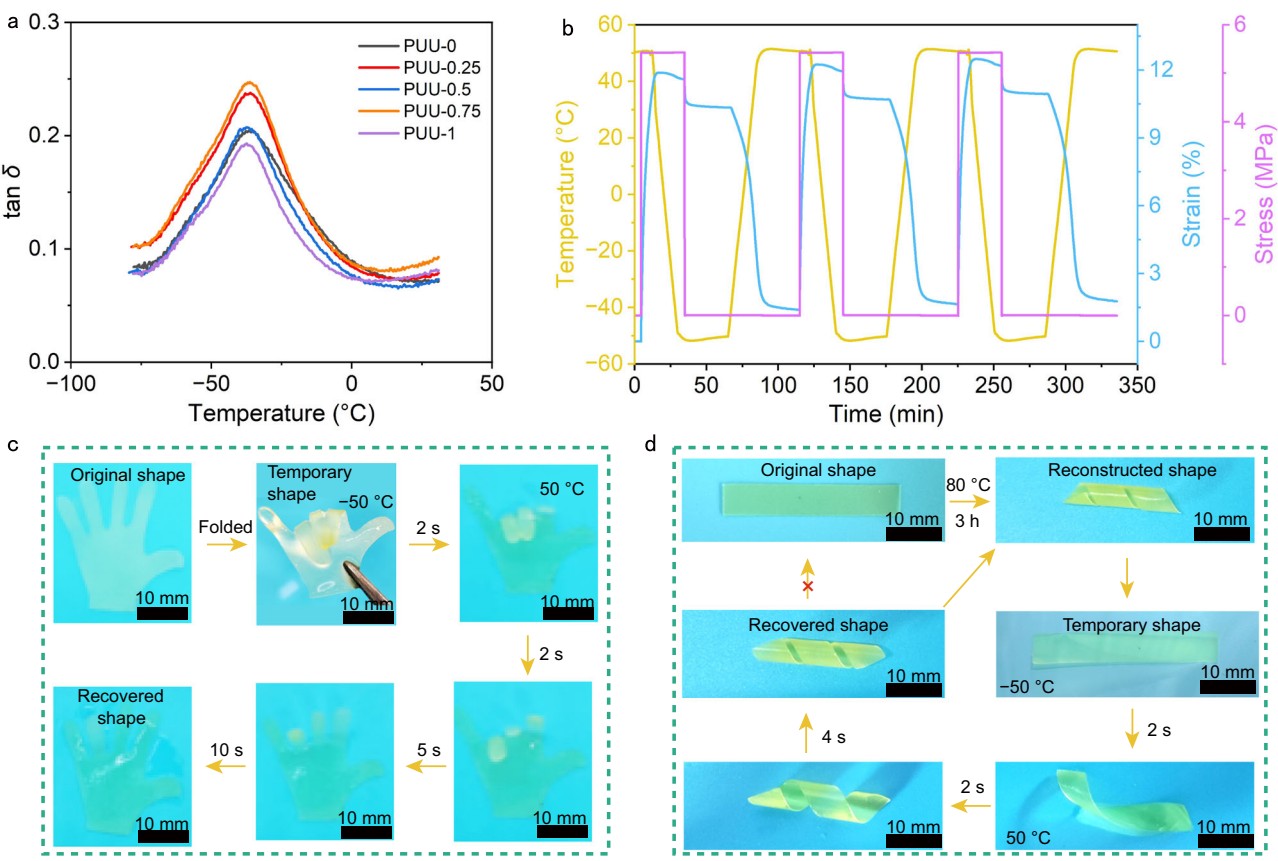

**Fig. 5 | Shape-memory and shape-reconfiguration attributes of PUU-0.5. a** Tan $\delta$ profiles of the PUU-X samples. **b** Shape-memory cycling curves of PUU-0.5. **c, d** Photographs demonstrating the (**c**) shape-memory and (**d**) shape-reconstructing properties of PUU-0.5.

remolded when the molecular chains were untangled and exchanged owing to the existence of dynamic supramolecular interactions. Supplementary Fig. 17 presents the stress-relaxation curves of PUU-0.5 at different temperature. We observed that the rate of stress-relaxation is faster with the increase of temperature, so we chose 80 °C as the temperature for shape reconstruction because it effectively promotes bond exchange and topological rearrangement and complete shape reconstruction in less time. Subsequently, a rectangular and tiled PUU-0.5 specimen was reconstructed into a spring shape at 80 °C for 3 h, and the material was fully unfolded and maintained at −50 °C thereafter to freeze the temporary configuration (Fig. 5d). Subsequently, the reconstructed

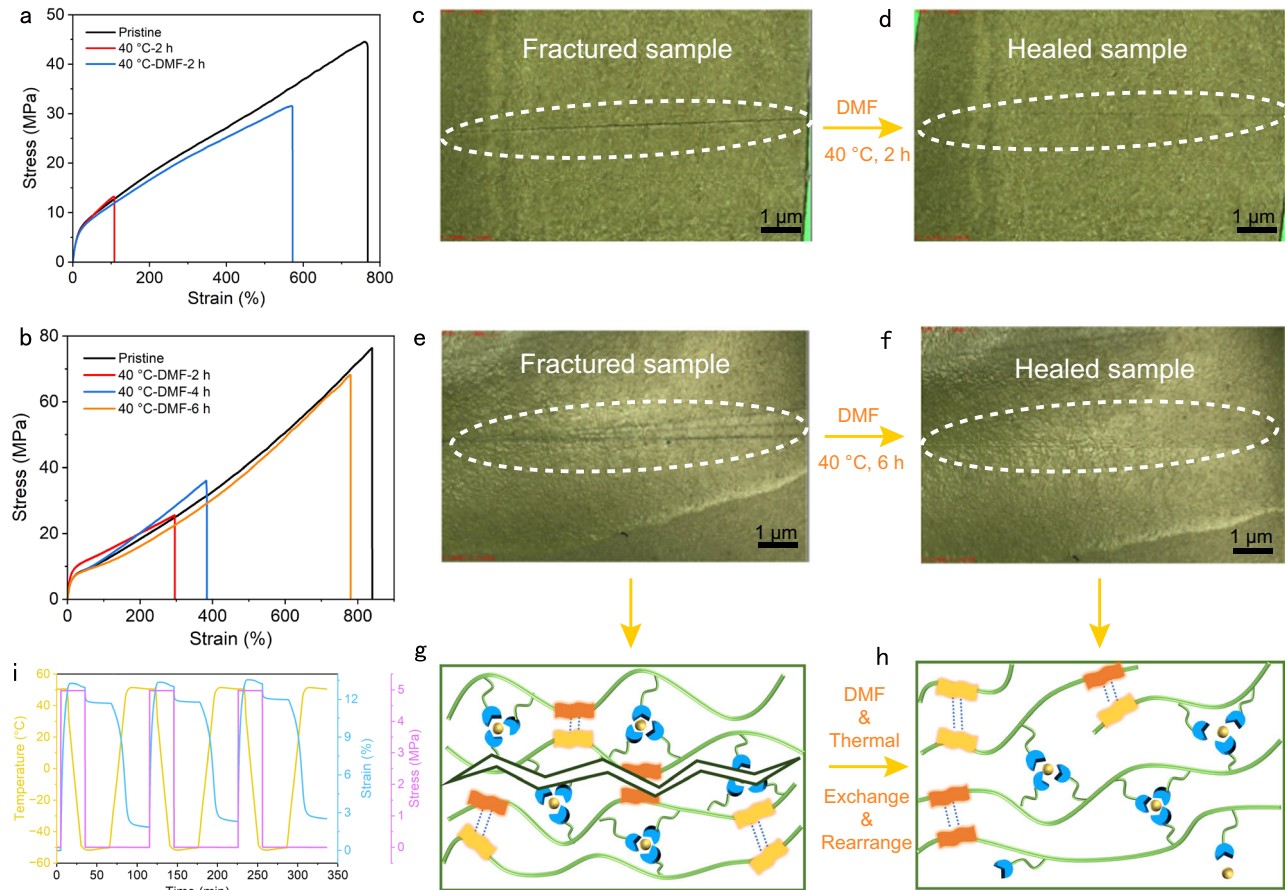

**Fig. 6 | Assessment of self-healing behavior. a** Tensile curves of pristine PUU-0, PUU-0 healed at 40 °C for 2 h, and PUU-0 healed at 40 °C for 2 h with DMF. **b** Tensile curves of pristine PUU-0.5 and those of PUU-0.5 healed at 40 °C for 2, 4, and 6 h with DMF. **c**, **d** Optical microscopic images of the fractured PUU-0.5 and PUU-0.5 healing at 40 °C for 2 h with DMF. **e**, **f** Optical microscopic images of the fractured PUU-0.5 and PUU-0.5 healing at 40 °C for 6 h with DMF. **g**, **h** Mechanism underlying the self-healing behavior of PUU-0.5. **i** Shape-memory cycling curves of healed PUU-0.5.

shape recovered to the spring shape rather than the initial rectangular configuration at 50 °C, indicating that PUU-0.5 exhibited shape-reconfigurable characteristics.

Self-healing tests. Dynamic reversible coordination and hydrogen bonds endowed the synthesized materials with self-healing properties. Therefore, the self-healing properties of PUU-0.5—the specimen that exhibited optimal mechanical properties—were systematically examined. The simple thermally induced healing strategy was compared with another approach, which involved adding solvent drops to the fracture surface while it underwent heating; this was done to reducing the healing duration by accelerating the molecular chain motion. First, PUU-0 was healed at 40 °C for 2 h with and without N,N-dimethylformamide (DMF, a solvent used in synthetic materials). The solvent-free healed samples exhibited a poor strength (Fig. 6a), resulting in a healing efficiency of only 30.31% (Eq. 3). In contrast, the solvent-containing samples exhibited a higher healing efficiency (69.41%). Consequently, drops of DMF were added to a crack on PUU-0.5, and the sample was incubated at 40 °C for different durations. Analysis of the tensile strengths of the samples (Fig. 6b) indicated that the material under the action of the solvent achieved a higher healing efficiency than that of the specimen healed using the simple thermal approach. Furthermore, the crack remained after the sample was healed at 40 °C for 2 h with DMF (Fig. 6c, d). Nevertheless, the healing efficiency gradually increased with prolonged duration and reached 91.72% after 6 h, following which the crack almost disappeared (Fig. 6e, f). Owing to the presence of heat and the solvent, the molecular chain motion and the nondynamic covalent bond dissociation and recombination were

accelerated (Fig. 6g, h), leading to crack repair. Analysis of the shape-memory properties of the healed PUU-0.5 (Fig. 6i) indicated that the sample retained its original performance; moreover, the corresponding $R_f$ (88.31%) and $R_r$ (96.27%) values were almost identical to those of the original material. These results suggest that the healed PUU-0.5 retained its notable shape-memory performance.

Shape-memory-assisted self-healing (SMASH). Wool and O'Connor[46] proposed a theory of crack healing in polymers, which includes (a) surface rearrangement, (b) surface approach, (c) wetting, (d) diffusion, and (e) randomization. Essentially, to heal cracks, two mechanically separated surfaces must physically be in contact with each other, which leads to diffusion of the molecular chains at the fracture surface. After the reactive groups are combined, the ruptured molecular chain segments are randomly connected, resulting in crack closure and repair. The force generated by the shape-memory polymer during shape recovery can act as the impetus to repair the fractured surface, causing the cracks to close. Additionally, because the temperature at which the temporary shape returns to its original shape is above $T_g$, the molecular chains begin to move actively at that temperature, thereby facilitating molecular chain penetration and entanglement at the fracture surface. Based on this, the shape-memory-driven self-healing effect of PUU-0.5 was scrutinized. Prior to these experiments, the conditions for enabling self-healing were explored in the last section and then applied to examine the shape-memory-driven self-healing. To that end, a temporarily bent PUU-0.5 specimen that was not completely fractured was heat-treated at 40 °C (Fig. 7a–c; 7a, side view; 7b, front view). The fractured surfaces were found to

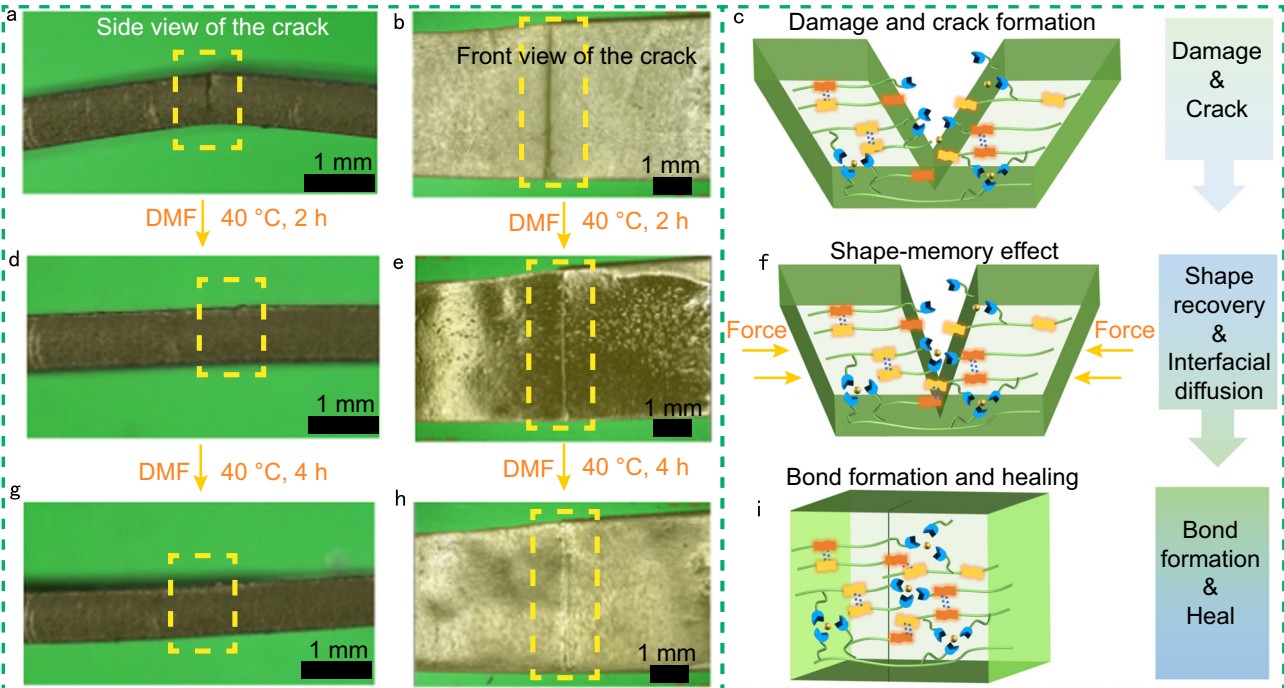

**Fig. 7 | SMASH attributes of PUU-0.5. a** Side and **b** front views of PUU-0.5. **c** Schematic of the fractured surfaces of the damaged sample. **d** Side and **e** front views of PUU-0.5 after healing for 2 h. **f** Illustration of progressive contact between the fractured surfaces of the damaged sample guided by the shape-memory effect. **g** Side and **h** front views of PUU-0.5 after healing for 4 h. **i** Schematic of the damaged sample being healed.

establish contact under the influence of the restoring force generated by the shape-memory effect, and a solvent was added dropwise onto the fractured surface thereafter. The length of the crack was significantly reduced after 2 h (Fig. 7d, e) because of the shape-recovering force that caused the molecular chains to recombine through the reversible interactions between coordination and hydrogen bonds after the fractured surfaces established contact (Fig. 7f). Furthermore, the crack almost disappeared after 4 h (Fig. 7g–i). These results suggest that the driving force of the shape-memory effect facilitated interfacial contact to permit self-healing.

CTE of PUU-X. Fluorescence was accidentally observed when all the PUU-X samples were irradiated with UV light. The optimal excitation and emission wavelengths of the materials ($\lambda_{ex}$ and $\lambda_{em}$, respectively) were determined by fluorescence spectrophotometry. All the materials could be excited at 354 nm (see Supplementary Fig. 18) owing to the presence of clusters such as urethane[47,48], urea[49,50], and carbonate[51]. These clusters further aggregated under the influence of hydrogen bonds, and strong nonbonding interactions occurred between the luminescent clusters in the material, resulting in through-space conjugation and a narrow bandgap. Furthermore, intramolecular motions such as rotations and vibrations were restricted, which led to the material emitting fluorescence upon absorbing UV light. To validate the existence of this phenomenon, the fluorescence intensity of the material was recorded at different temperatures (Fig. 8b, d). The results indicated that the fluorescence intensity decreased gradually with increasing temperature; this was due to the luminescent groups no longer clustering at high temperatures owing to the dynamic bond dissociation. Therefore, the luminescence of the material was confirmed to be associated with dynamic bonds. Additionally, PUU-X exhibited a $Zn^{2+}$-content-dependent CTE (Fig. 8a). The highest luminescence intensity among the specimens—exhibited by PUU-0—decreased gradually with increasing $Zn^{2+}$ content owing to the commensurate enhancement in fluorescence quenching[52]. Concurrently, the addition of zinc ions increased the strength of the conjugated system, leading to bandgap narrowing and a redshift in $\lambda_{em}$. Overall, PUU-0.5 exhibited better comprehensive performance metrics than those of numerous similar specimens reported previously[30,48,53–57] (Fig. 8c).

In summary, a multifunctional elastomer with good mechanical, SMASH, and CTE properties was successfully synthesized. Tuning the coordination interactions and utilizing their synergy with hydrogen bonds endowed PUU-0.5 with ultrahigh strength (76.37 MPa) and noteworthy elongation at break (839.10%) and toughness (308.63 MJ m$^{-3}$). Owing to the reversibility of the supramolecular interactions, the PUU-X specimens could maintain their temporary shape and then return to their original shape after heating. The damaged surfaces were brought closer under the influence of the shape recovery impetus, which promoted molecular chain recombination owing to the reversible interactions between coordination and hydrogen bonds; furthermore, cracks healed under mild conditions with the assistance of DMF. Finally, PUU-X exhibited CTE properties, with all samples emitting cyan fluorescence at 354 nm. Overall, a strategy for developing multifunctional high-performance materials was implemented in this study.

## Methods

Materials. PCDL ($M_n$ = 2000 Da, technical grade) was acquired from Hongming Chemical Reagents (Jining). HDI (purity: 99%), Zn(OTf)$_2$ (purity: 98%), and SDH (purity: 99%) were sourced from Energy Chemical Technology (Shanghai). BIDI (purity: ≥98%) was purchased from Aladdin (Shanghai, China). Ethanol (analytically pure), dimethyl sulfoxide (DMSO, analytically pure), and DMF (analytically pure) were purchased from Rionlon Bohua (Tianjin) Pharmaceuticals & Chemicals. Dibutyltin dilaurate (DBTDL, analytically pure) was obtained from Tianjin No. 1 Chemical Reagent Factory. PCDL is dried to removal water before use, and other reagents are used directly without further purification.

Synthesis of PUU-X. PCDL ($M_n$ = 2000 Da, 2.5 g, 1.25 mmol) was initially dried in a vacuum oven at 120 °C for 3 h. DMF (20 mL) and

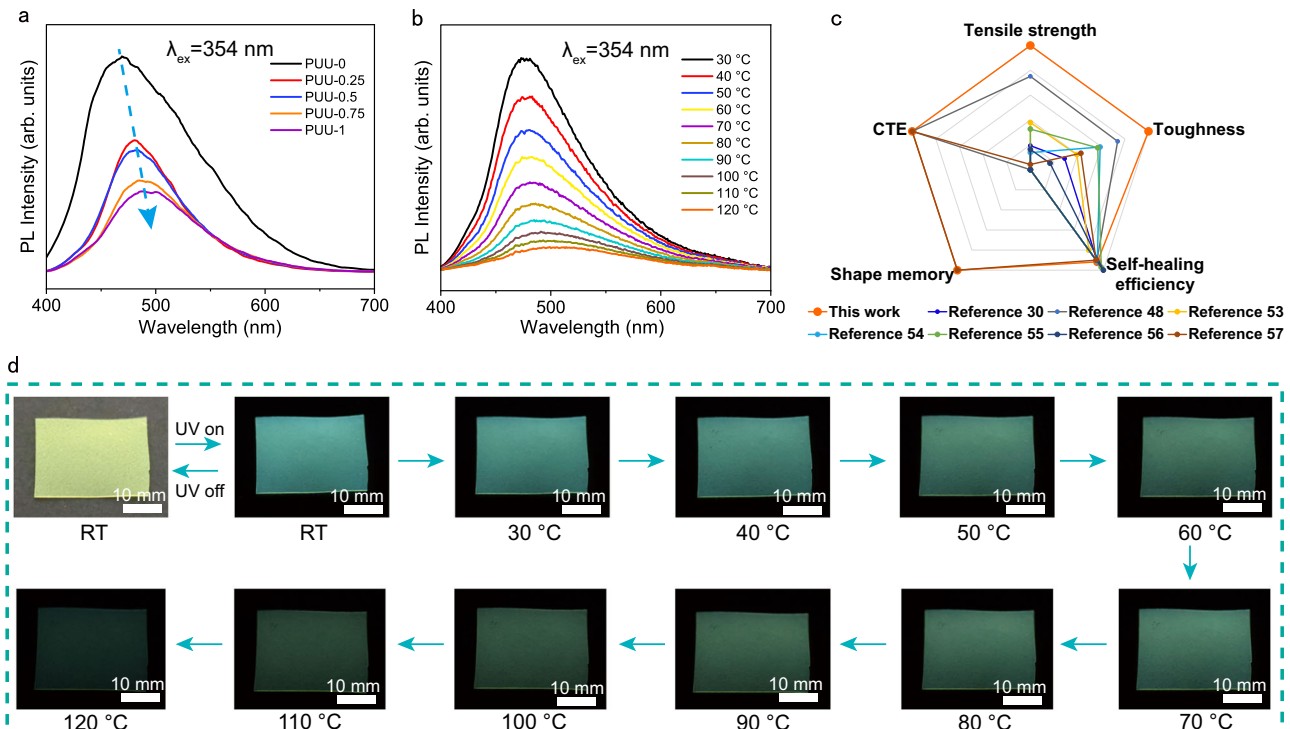

**Fig. 8 | CTE from PUU-X. a, b** Photoluminescence spectra of (**a**) PUU-X specimens with different $Zn^{2+}$ contents (**b**) and PUU-0.5 at different temperatures. **c** The comparison between this work and the reported work in terms of comprehensive performance after normalization in terms of mechanical, healing, and emission characteristics. **d** Photographs of PUU-0.5 acquired at different temperatures under a UV lamp.

DBTDL (50 µL, 0.083 mmol) were added to a three-necked flask containing PCDL and stirred thoroughly in a $N_2$ atmosphere. Next, a predetermined amount of HDI (0.4 mL, 2.5 mmol) was added dropwise to the three-necked flask and reacted for 3 h. SDH (0.37 g, 2.5 mmol) dissolved in DMSO (50 mL) was added dropwise thereafter. After 2 h, BIDI (0.27 g, 1.25 mmol) dissolved in DMSO (10 mL) was added to the system and reacted for 1 h. Appropriate solvents were added during the reaction to reduce viscosity, and the aforementioned procedures were performed at 60 °C. $Zn(OTf)_2$ (0.076 g, 0.2 mmol) dispersed in DMF (5 mL) was subsequently added dropwise into the flask. The mucus-like material obtained after 30 min was poured into a mold and placed in an oven at 60 °C for solvent removal, subsequently yielding a light-yellow film; this sample is denoted as PUU-0.5. PUU-X specimens (where X is the $Zn^{2+}$/pyridine molar ratio) were similarly prepared using different molar ratios of $Zn(OTf)_2$/pyridine. The molar ratios of the functional groups of the monomers are listed in Supplementary Table 1.

Characterization. FTIR spectroscopy (Nicolet Nexus 870 spectrometer, Bruker, Germany) was performed to characterize PUU-X in the attenuated total reflection mode from 500 to 4000 $cm^{-1}$ at room temperature. Solid-state $^{13}$C NMR spectra were recorded using an AVANCE NEO 400 M spectrometer (Bruker, Germany). A thermogravimetric analyzer (STA449F3, Germany) was used to determine the thermal stability of PUU-X, with all samples heated from room temperature to 600 °C at a rate of 10 °C $min^{-1}$ in a $N_2$ atmosphere. Thermomechanical properties and stress-relaxation were characterized using a dynamic mechanical analyzer (NETZSCH DMA 242 C). To that end, all PUU-X specimens were cut into rectangles (20 × 3 × 0.1–0.3 $mm^3$) and heated from −80 to 30 °C at a rate of 5 °C $min^{-1}$ under a 1 N load and 1 Hz frequency. An electronic universal tensile testing machine (AG-X, Shimadzu, Japan) was used to analyze the mechanical properties of PUU-X. For these experiments, all samples were cut into dumbbell shapes (30 × 2 × 0.1–0.3 $mm^3$) and tested at room temperature at a tensile rate of 10 mm $min^{-1}$. SAXS patterns were acquired

using a Nanostar U SAXS device (Bruker, Germany) equipped with a multilayer focused Cu Kα X-ray source (IµS 30 W, Incoatec; λ = 0.154 nm). The morphology of PUU-X was examined by performing XRD (EMPYREAN) in a scan range of 10–90° with Cu Kα radiation. A fluorescence spectrophotometer (Omni-λ300i, Zolix Instruments) was used to obtain the photoluminescence spectra. In situ fluorescence spectra (for temperatures of 30–120 °C) were collected using an FLS920 fluorescence spectrometer (Edinburgh Instruments). UV–vis absorption spectra were obtained using a UV–vis near-infrared diffuse reflection instrument (Shimadzu UV-3600, Japan). In situ temperature-dependent FTIR spectroscopy (at 30–120 °C) was performed using an IRAffinity-1S spectrometer (Shimadzu, Japan) with a resolution of 4 $cm^{-1}$ by acquiring 32 scans from 4000 to 600 $cm^{-1}$.

Shape-memory tests. The shape-memory properties of PUU-X were scrutinized by DMA (NETZSCH DMA 242 C). Each sample was cut into rectangles (20 × 3 × 0.1–0.3 $mm^3$) and heated at 50 °C. The sample was subjected to an original strain (denoted as $\varepsilon_A$) and then stretched and maintained for 5 min. After noting the resulting length (denoted as $\varepsilon_B$), the sample was cooled to −50 °C and maintained for 30 min. Subsequently, a temporary shape ($\varepsilon_B$) was obtained after the force was removed. $\varepsilon_{A,rec}$ was determined after heating the sample to 50 °C and maintaining it for 30 min.

$$R_f(A \rightarrow B) = \frac{\varepsilon_B - \varepsilon_A}{\varepsilon_{B,load} - \varepsilon_A} \times 100\% \tag{1}$$

$$R_r(B \rightarrow A) = \frac{\varepsilon_B - \varepsilon_{A,rec}}{\varepsilon_B - \varepsilon_A} \times 100\% \tag{2}$$

Self-healing tests. The ability of the synthesized materials to crack and heal was probed under different conditions. The tensile strengths of the cracked and healed samples ($\sigma_0$ and $\sigma$, respectively) were used to

calculate the healing efficiency ($\eta$) as follows:

$$\eta = \frac{\sigma}{\sigma_0} \times 100\% \qquad (3)$$

SMASH tests. A crack was created when the sample assumed the temporary configuration. Then, the crack started to close upon exposing the sample to 40 °C. Subsequently, DMF was added to the crack to promote the molecular chain motion and accelerate healing. The crack morphology was examined using a video-measuring system (Rational WANHAO, VMS-2515H).

## Data availability
The raw data generated in this study have been deposited in the figshare database under accession code [https://doi.org/10.6084/m9.figshare.23499474], and all other data are available from the author upon request.

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

## Acknowledgements

This study was financially supported by the CAS Project for Young Scientists in Basic Research (YSBR-023, X.Z.), the Youth Innovation Promotion Association of the Chinese Academy of Sciences (Y2022103, X.Z.), the National Natural Science Foundation of China (Nos.51935012, Q.W. 52005481, Z.Y. and 52205234, S.L.), and the Key Program of the Lanzhou Institute of Chemical Physics, CAS (KJZLZD-3, X.Z. and ZYFZFX-7, Y.Z.).

## Author contributions

X.W., S.L., Z.Y., and X.Z. provided experimental ideas, wrote the manuscript and materials synthesis. J.X. and Y.Z. participated in the design of figures. T.W. and Q.W. revised the first draft.

## Competing interests

The authors declare no competing interests.
