## [Peer Review File · Nature Communications]

A stretchable, mechanically robust polymer exhibiting shape-memory-assisted self-healing and clustering-triggered emissionREVIEWER COMMENTS

Reviewer #1 (Remarks to the Author):

This manuscript proposed a strategy for constructing a multifunctional elastomer with outstanding mechanical properties, shape memory assisted self-healing, and clusterization-triggered emission (CTE) properties by tuning the coordination interaction, hydrogen bonds and the synergistic effect of both. Impressively, the PUU-0.5 possess excellent mechanical properties with an ultrahigh strength of 76.37 MPa, elongation at-break of 839.10%, and toughness of 308.63 MJ m⁻³. It is worth noting that the self-healing mechanism of shape memory polymer in this work is realized by the combined effect of driving force of shape memory, interaction between hydrogen bonds and coordination bonds, and the activation of molecular chains by solvents, which is different from traditional self-healing strategies (Nature,2016,540(7633):363-370).It is of great interest for researchers working in the field of cryptographic material, intelligent detection, sealing protection, responsive polymer materials and so on. I think this work can be published on top journal as Nature Communications. However, some minor issues need to be solved and some parts of the text should be further improved before publication.

1. Your manuscript requires revision with respect to the language used. I suggest that you ask a native English speaker or equivalent to assist you with correcting the spelling, grammar, word use, and punctuation throughout your manuscript.
2. Some minor details need to be revised, for example, "coordination bonds" should be capitalized on line 5 of page 2, The legend of Fig. 7 is incorrectly marked (C is written as A). The "PUUS" on legend of Fig. S5 need to be corrected
3. I think the last sentence of the abstract (This multifunctional material with outstanding mechanical properties is expected to be used in harsh working conditions) is awkward, and the first sentence on line 2 of page 10 (The cracks were cut out of the sample when it was in a temporary shape) is misstated, please revise it.
4. The structure characterization of all PUUs is insufficient. NMR characterization and relevant discussions should be provided.
5. The paper emphasizes the good mechanical properties of the materials, which I think should be compared with the work reported so far.
6. Please determine whether the parentheses on line 11 of page 25 are content of the

manuscript.

7. It is suggested that related references about supramolecular PU be cited in the manuscript, such as "Nat Commun 13, 4868 (2022)", "Nat Commun 12, 1291 (2021)".

Reviewer #2 (Remarks to the Author):

In the paper titled "Mechanically robust and stretchable polymer with shape memory assisted self-healing and clusterization-triggered emission", the authors reported an effective way to prepare multi-functional polyurethane-based elastomers. It is noteworthy that the ionic interaction and plenty of hydrogen bonds endow elastomer with impressive strength, stretchability, and toughness simultaneously which provides a promising candidate to be used for engineering elastomer and high-performance elastic matrix. Overall, this work examined the potential value of polyurethane elastomer in depth in terms of mechanical behaviors, self-healing and fluorescence emission. However, there are still some concerns that need to be addressed and the overall writing needs to be further polished. I recommend publication of the manuscript after the following points have been addressed:

1. In Figure 1A, page 12, the authors explained their design for synthesizing the polyurethane-urea (PUU) supramolecular elastomer. In the third step, the authors utilized the reaction between amine groups on the end of polyurethane precursor chains and aldehyde groups to form imine groups, and thus introduce the pyridine for ionic coordination. However, it looks like there is a mistake in the chemical structure of the final products, the authors labeled the double bond in the wrong position, it should be "C=N" not "C=C". Please check and revise. Additionally, the reaction between amine and aldehyde will be affected by the acidic or basic condition, please specify the acidic or basic condition in the experimental part.

2. In line 215, page 13. the authors' hypothesis for peak shift is correct. For FTIR, the vibrational modes are very susceptible to the formation of hydrogen bonds because hydrogen bonds significantly alter the electronic structure of the molecular system and, consequently, their associated vibrational transitions. The peak will have a red shift when the corresponding groups are hydrogen bonded. Upon increasing temperature, the degree

of hydrogen bonding will decrease, which means there will be more free groups generated, so the peak will shift to the left. Please review and revise, may cite this paper (<https://doi.org/10.1021/acs.jchemed.5b01014>).

3. In line 209, page 13, the authors claimed “Nevertheless, the mechanical properties of PUU-X declined following the augmentation of Zn²⁺, which may be due to the agglomeration of physical cross-linking and consequent stress concentration,” and in line 392, page 25, the authors claimed “The luminescence intensity of the material decreases gradually with increasing Zn²⁺ content due to the addition of zinc ions resulting in fluorescence quenching (fluorescent metallopolymer with Zn (II) in a Schiff base/phenoxide coordination environment). Moreover, the addition of zinc ions increases the conjugated system resulting in narrows the band gap and redshifts λ_{em} .” Please cite the relevant references to support those explanations.

4. In this work, the authors introduced plenty of non-covalent bonds into the polymer matrix and there was no chemical crosslinking in the synthesis of PUU, which means the polymer network is only physical-crosslinked by dynamic supramolecular interaction. Is it possible to realize the recyclability of PUU or reconfiguration of shape memory process?

5. Since Zn²⁺ ions were introduced into PUU and the special pyridine motif inside PUU, wondering if the conductivity (or resistance) of PUU will be affected.

6. Please further polish the writing and clarify parts that are read vaguely. For instance, in line 39, page 3, the authors claimed “However, conventional PUs have gradually failed to meet people’s needs, so ingenious molecular designs are needed to obtain multifunctional and novel materials,”. It was not clear what people’s needs the authors refer here.

Reviewer #3 (Remarks to the Author):

The authors develop a new type of supramolecular elastomer material that can maintain excellent mechanical properties while possessing shape memory and self-healing functions, as well as clusterization-triggered emission effect. This tough multifunctional material is expected to be suitable for harsh working environments. In this manuscript, the authors introduce a new synthesis method that introduces metal coordination bonds. It retains excellent mechanical properties while possessing self-healing and shape memory properties. The authors conducted a large number of experiments as well as accurate

analysis. In addition, the authors combined experiments with simulations, which added credibility to this work. However, several points need substantial improvement. My comments are :

Major issues:

1. In the manuscript, a large amount of testing and analysis were conducted on the mechanical properties of the materials. Whether the multifunctional properties of materials has been ignored? As the novelty of this work is the combination of mechanical properties and versatility multifunctional properties. There is a lack of principled explanation for self-healing and shape memory properties.
2. An important aspect of this work is the addition of $Zn(OTf)_2$. However, there is a lack of systematic discussion on its function and necessity in this manuscript. Two-dimensional infrared correlation spectrum is an interesting tool for analyzing dynamic bond changes. You have analyzed hydrogen bonds, can you also do the same for metal coordination bonds?

Detail issues:

1. It would be useful to include some comparison charts between this work and other works.
2. Annotations on illustrations can provide more information.
3. In Figure 3(as well as Figure 1), there should be two hydrogen bonds between two SDH molecules(D1), as is showed in D2.
4. It is better to provide a locally enlarged image of Figure 2D and Figure 2E.
5. In table 1, "stress" and "strain" should be "Tensile strength" and "strain at break".
6. Line 97 and Line 101: It's better to give the mole numbers of BIDI and DBTDL.
7. Line 148: " ϵ_B " should be " $\epsilon_{B,load}$ ".
8. Line 193: "FTTR" should be "FTIR". It is better to provide a locally enlarged image of Figure 2D and Figure 2E.
9. Line 198: It is best not to use "poor stress", as it still have good mechanical properties. It is better to use "worst tensile strength".

Dear reviewers,

We are grateful for your consideration of our manuscript “*Mechanically robust and stretchable polymer with shape memory assisted self-healing and clusterization-triggered emission*” (the revised title is “*A stretchable, mechanically robust polymer exhibiting shape-memory-assisted self-healing and clustering-triggered emission*”), and we would like to thank all reviewers for constructive comments and suggestions which have been very helpful in improving the manuscript. After carefully reading and thinking about reviewers’ comments, we have made supplemented this article and performed incorporated changes to reflect most of the suggestions provided by the reviewers. We have supplemented characterizations, added discussions, revised content and figures, and we have corrected the spelling, grammar, word use, and punctuation throughout our manuscript with the help of native English speaker. All corrections were in red font and supplementary contents were in blue font in the revised manuscript. In any case, we are open to consideration of any further comments and suggestions.

Sincerely,

Song Li, Zenghui Yang and Xinrui Zhang

Followings are the responses to reviewers’ comments point to point,

Point-by-point response to the reviewers’ comments

Reviewer #1 (Remarks to the Author):

This manuscript proposed a strategy for constructing a multifunctional elastomer with outstanding mechanical properties, shape memory assisted self-healing, and clusterization-triggered emission (CTE) properties by tuning the coordination interaction, hydrogen bonds and the synergistic effect of both. Impressively, the PUU-0.5 possess excellent mechanical properties with an ultrahigh strength of 76.37 MPa, elongation at-break of 839.10%, and toughness of 308.63 MJ m⁻³. It is worth noting

that the self-healing mechanism of shape memory polymer in this work is realized by the combined effect of driving force of shape memory, interaction between hydrogen bonds and coordination bonds, and the activation of molecular chains by solvents, which is different from traditional self-healing strategies (Nature,2016,540(7633):363-370).It is of great interest for researchers working in the field of cryptographic material, intelligent detection, sealing protection, responsive polymer materials and so on. I think this work can be published on top journal as Nature Communications. However, some minor issues need to be solved and some parts of the text should be further improved before publication.

Response: Thank you for your insightful and constructive comments. To address your and other reviewers' comments and concerns in full, we have added or revised figures, expanded discussions and improved the language in the revised manuscript. The main changes in the revised manuscript or supplementary information are in red (revised contents) and blue (supplementary contents). In the following paragraphs, we will address each comment point by point.

1. Your manuscript requires revision with respect to the language used. I suggest that you ask a native English speaker or equivalent to assist you with correcting the spelling, grammar, word use, and punctuation throughout your manuscript.

Response: Thank you for your valuable comment. We have corrected the spelling, grammar, word use, and punctuation throughout our manuscript with the help of a native English speaker. After polishing, the article reads more smoothly and expresses more clearly. Thank you for your comment.

2. Some minor details need to be revised, for example, “coordination bonds” should be capitalized on line 5 of page 2, The legend of Fig. 7 is incorrectly marked (C is written as A). The “PUUS” on legend of Fig. S5 need to be corrected.

Response: Thank you for your careful review of the manuscript and your comments. We apologize for the error of spell and legend, and we've corrected it in the revised manuscript and Supplementary Information.

3. I think the last sentence of the abstract (This multifunctional material with outstanding mechanical properties is expected to be used in harsh working conditions) is awkward, and the first sentence on line 2 of page 10 (The cracks were cut out of the sample when it was in a temporary shape) is misstated, please revise it.

Response: Thank you for your valuable suggestion. We have revised the last sentence of the abstract (The strategy reported herein for developing multifunctional materials with outstanding mechanical properties can be leveraged to yield stimulus-responsive polymers and smart seals.) and corrected the sentence on page 10 is now line 1, page 27 (A crack was created when the sample assumed the temporary configuration.).

4. The structure characterization of all PUUs is insufficient. NMR characterization and relevant discussions should be provided.

Response: Thanks for your valuable comment. According to your advice, the solid state ¹³C NMR characterizations have been added in the revised **Supplementary Information**, as shown in **Figs. R1-R5** (**Figs. S2-S6** in Supplementary Information). From the results of NMR further demonstrated that the materials have been synthesized successfully.

Fig. R1 (Fig. S2 in Supplementary Information) The solid state ^{13}C NMR of PUU-0.

Fig. R2 (Fig. S3 in Supplementary Information) The solid state ^{13}C NMR of PUU-0.25.

Fig. R3 (Fig. S4 in Supplementary Information) The solid state ^{13}C NMR of PUU-0.5.

Fig. R4 (Fig. S5 in Supplementary Information) The solid state ^{13}C NMR of PUU-0.75.

Fig. R5 (Fig. S6 in Supplementary Information) The solid state ^{13}C NMR of PUU-1.

5. The paper emphasizes the good mechanical properties of the materials, which I think should be compared with the work reported so far.

Response: Thank you for your valuable suggestion. According to your advice, we searched relevant works for comparison of mechanical properties, and drew a three-dimensional diagram, as shown in the **Fig. R6 (Fig. 2c in revised manuscript)** below. In addition to this, we also compared the comprehensive properties of this material with other reported works. As shown in the **Fig. R7 (Fig. 8c in revised manuscript)**, the material showed good comprehensive properties.

Fig. R6 (Fig. 2c in revised manuscript) Comparison between PUU-0.5 and similar previously reported specimens in terms of strength, strain, and toughness.

Fig. R7 (Fig.8c in revised manuscript) Superiority of PUU-X over similar previously reported samples after normalization in terms of mechanical, healing, and emission characteristics.

6. Please determine whether the parentheses on line 11 of page 25 are content of the manuscript.

Response: Thank you for your careful review of the manuscript. We apologize for this error of reference format. In fact, this is the cited reference, this part has been deleted

and the reference has been correctly cited (The highest luminescence intensity among the specimens—exhibited by PUU-0—decreased gradually with increasing Zn^{2+} content owing to the commensurate enhancement in fluorescence quenching⁵².)

7. It is suggested that related references about supramolecular PU be cited in the manuscript, such as “Nat Commun 13, 4868 (2022)”, “Nat Commun 12, 1291 (2021)”.

Response: We thank the reviewer for the important criticism, and we've cited relevant references on “For instance, some PU block copolymers such as PUU and other materials with diverse functions such as self-healing, shape-memory, and recycling characteristics have recently been developed through skillful molecular design¹⁴⁻²¹.”, see page 3 of the manuscript, line 5.

Reviewer #2 (Remarks to the Author):

In the paper titled “Mechanically robust and stretchable polymer with shape memory assisted self-healing and clusterization-triggered emission”, the authors reported an effective way to prepare multi-functional polyurethane-based elastomers. It is noteworthy that the ionic interaction and plenty of hydrogen bonds endow elastomer with impressive strength, stretchability, and toughness simultaneously which provides a promising candidate to be used for engineering elastomer and high-performance elastic matrix. Overall, this work examined the potential value of polyurethane elastomer in depth in terms of mechanical behaviors, self-healing and fluorescence emission. However, there are still some concerns that need to be addressed and the overall writing needs to be further polished. I recommend publication of the manuscript after the following points have been addressed:

Response: Many thanks for your positive evaluation for novelty and results of our work. We have added related characterizations, revised images, and discussed more in-depth based on your suggestions and comments. In addition, we have further polished the writing and clarify parts with help of native English speaker. The main changes in the revised manuscript or supporting information are in red (revised contents) and blue (supplementary contents). In the following text, we will address each comment point by point.

1. In Figure 1A, page 12, the authors explained their design for synthesizing the polyurethane-urea (PUU) supramolecular elastomer. In the third step, the authors utilized the reaction between amine groups on the end of polyurethane precursor chains and aldehyde groups to form imine groups, and thus introduce the pyridine for ionic coordination. However, it looks like there is a mistake in the chemical structure of the final products, the authors labeled the double bond in the wrong position, it should be “C=N” not “C=C”. Please check and revise. Additionally, the reaction between amine and aldehyde will be affected by the acidic or basic condition, please specify the acidic

or basic condition in the experimental part.

Response: Thank you for your careful review of the manuscript. We apologize for the error of chemical structure and we've corrected it in the revised manuscript and as shown in **Fig. R8**. For the reaction process, we didn't add an acid or a base to the process, and the FT-IR also showed that the amine reacted with the aldehyde group to form C=N (the peak of the acyl hydrazone bond (C=N) appeared at 1662 cm^{-1}).

Fig. R8 (Fig. 1a in revised manuscript) Schematic illustrating the synthesis of PUU-X elastomers.

2. In line 215, page 13. the authors' hypothesis for peak shift is correct. For FTIR, the vibrational modes are very susceptible to the formation of hydrogen bonds because hydrogen bonds significantly alter the electronic structure of the molecular system and, consequently, their associated vibrational transitions. The peak will have a red shift when the corresponding groups are hydrogen bonded. Upon increasing temperature, the degree of hydrogen bonding will decrease, which means there will be more free groups generated, so the peak will shift to the left. Please review and revise, may cite this paper (<https://doi.org/10.1021/acs.jchemed.5b01014>).

Response: Thanks for your construction suggestions. We have revised the section of in-situ FTIR, and relevant paper (<https://doi.org/10.1021/acs.jchemed.5b01014>) has been cited. The modified content is as follows.

Noteworthy changes were also observed in the acquired FTIR spectra (**Figs. R9-R11, Fig. S9 in Supplementary Information, Figs.2e, and 2f** in revised manuscript).

For instance, the C=O peak shifted from 1737 to 1742 cm^{-1} and enhanced in intensity with increasing temperature (**Fig. R10, Fig. 2e** in revised manuscript). Furthermore, the intensity of the peak at 1546 cm^{-1} (corresponding to C-N) enhanced with rising temperature (**Fig. R11, Fig. 2f** in revised manuscript). These trends were presumably due to the gradual dissociation of the coordination bonds when the material was heated and the increase in the number of pyridine groups released. Two-dimensional infrared correlation spectra (2D-COS) of PUU-0.5 (**Figs. R12 and R13, Figs. S10 and S11 in Supplementary Information**) were acquired to validate these results. The synchronous spectrum of the sample (**Fig. R12, Fig. S10 in Supplementary Information**) displayed four main autopeaks—(1747, 1747), (1734, 1734), (1712, 1712), and (1653, 1653)—whereas the asynchronous spectrum (**Figs. R13, Fig. S11 in Supplementary Information**) showed five main cross-peaks—(1747, 1734), (1747, 1653), (1734, 1712), (1734, 1653), and (1712, 1653). According to Noda's rule^{44, 45}, the temperature sensitivity of each peak during the heating, ordered from fast to slow, is 1747 cm^{-1} > 1734 cm^{-1} > 1712 cm^{-1} > 1653 cm^{-1} . The 1653 and 1712 cm^{-1} peaks mainly correspond to the C=N affected by coordination bonds and the ordered C=O influenced by hydrogen bonds, respectively. Moreover, the 1734 and 1747 cm^{-1} peaks are linked to the semi-ordered C=O partially affected by hydrogen bonding and the free C=O without hydrogen bonding, respectively. Considering the original infrared spectral data acquired in the 1600–1800 cm^{-1} band, the relative intensities of the absorption peaks at approximately 1710, 1747, and 1653 cm^{-1} increased steadily with increasing temperature, indicating that the hydrogen and coordination bonds in PUU-X were

continuously ruptured, thereby forming freer C=O, N-H, and C=N bonds. Overall, these results collectively validate the existence of hydrogen and coordination bonds in the material.

Fig. R9 (Fig. S9 in Supplementary Information) The *in-situ* temperature-dependent FTIR of PUU-0.5.

Fig. R10 (Fig. 2e in revised manuscript) Two-dimensional infrared (1755-1725 cm^{-1})

investigation of PUU-0.5.

Fig. R11 (Fig. 2f in revised manuscript) Two-dimensional infrared (1570-1530 cm⁻¹) investigation of PUU-0.5.

Fig. R12 (Fig. S10 in Supplementary Information) The synchronous spectrum of the

PUU-0.5.

Fig. R13 (Fig. S11 in Supplementary Information) The asynchronous spectrum of the PUU-0.5.

3. In line 209, page 13, the authors claimed “Nevertheless, the mechanical properties of PUU-X declined following the augmentation of Zn²⁺, which may be due to the agglomeration of physical cross-linking and consequent stress concentration,” and in line 392, page 25, the authors claimed “The luminescence intensity of the material decreases gradually with increasing Zn²⁺ content due to the addition of zinc ions resulting in fluorescence quenching (fluorescent metallopolymers with Zn (II) in a Schiff base/phenoxide coordination environment). Moreover, the addition of zinc ions increases the conjugated system resulting in narrows the band gap and redshifts λ_{em} .” Please cite the relevant references to support those explanations.

Response: Thank you for your valuable suggestion. About the explanation of mechanical properties, we found the reference (*Chinese Journal of Chemical*

Engineering 53 (2023): 211-221.) and explained the decline in mechanical properties. The paper has been cited in the revised manuscript. For the section of “The luminescence intensity of the material decreases gradually with increasing Zn²⁺ content due to the addition of zinc ions resulting in fluorescence quenching (fluorescent metallopolymers with Zn (II) in a Schiff base/phenoxide coordination environment). The content in brackets is the reference explaining the fluorescence quenching. We are sorry for the wrong reference format, and we have revised it. The modified content is as follows.

The highest luminescence intensity among the specimens—exhibited by PUU-0—decreased gradually with increasing Zn²⁺ content owing to the commensurate enhancement in fluorescence quenching⁵².

4. In this work, the authors introduced plenty of non-covalent bonds into the polymer matrix and there was no chemical crosslinking in the synthesis of PUU, which means the polymer network is only physical-crosslinked by dynamic supramolecular interaction. Is it possible to realize the recyclability of PUU or reconfiguration of shape memory process?

Response: Thank you for your valuable suggestion. For recyclability of PUU, we have done experiments, due to the existence of a variety of strong and weak physical crosslinking, so that the material cannot be completely dissolved in the solvent, so the material can not realize recyclability. For reconfiguration of shape memory process, we have added experiments and descriptions of it. **Fig. R14 (Fig. S17** in Supplementary Information) presents the stress-relaxation curves of PUU-0.5 at different temperature. We observed that the rate of stress-relaxation is faster with the increase of temperature, so we chose 80 °C as the temperature for shape reconstruction because it effectively promotes bond exchange and topological rearrangement and complete shape reconstruction in less time. Subsequently, a rectangular and tiled PUU-0.5 specimen was reconstructed into a spring shape at 80 °C for 3 h, and the material was fully unfolded and maintained at -50 °C thereafter to freeze the temporary configuration (**Fig. R15, Fig. 5d** in revised manuscript). Subsequently, the reconstructed shape recovered

to the spring shape rather than the initial rectangular configuration at 50 °C, indicating that PUU-0.5 exhibited shape-reconfigurable characteristics.

Fig. R14 (Fig. S17 in Supplementary Information). The stress-relaxation curves of PUU-0.5 at different temperature.

Fig. R15 (Fig. 5d in revised manuscript). Shape-reconstructing properties of PUU-0.5.

5. Since Zn^{2+} ions were introduced into PUU and the special pyridine motif inside PUU, wondering if the conductivity (or resistance) of PUU will be affected.

Response: We thank the reviewer for the important criticism. According to your

question, we have tested the conductivity and resistivity of PUU-X, and the results are shown in the table below. The test results showed that all the PUU-X barely conducted electricity.

Table R1 The conductivity and resistivity of PUU-X.

Samples	PUU-0	PUU-0.25	PUU-0.5	PUU-0.75	PUU-1
Conductivity	12.71 ±	657.57 ±	251.33 ±	400.77 ±	381.43 ±
(S/cm)	0.41T	35.06G	12.38G	33.54G	50.04G
Resistivity	0.079 ±	0.0015 ±	0.0040 ±	0.0025 ±	0.0027 ±
(Ω·cm)	0.0025p	7.63E-5p	1.95E-4p	2.1E-4p	3.37E-4p

T=10¹², G=10⁹, p=10⁻¹²

6. Please further polish the writing and clarify parts that are read vaguely. For instance, in line 39, page 3, the authors claimed “However, conventional PUs have gradually failed to meet people’s needs, so ingenious molecular designs are needed to obtain multifunctional and novel materials,”. It was not clear what people’s needs the authors refer here.

Response: Thank you for your valuable suggestion. There are indeed some problems in the writing and grammar of the manuscript. According to your suggestions, we have polished the full text. After polishing, the article reads more smoothly and expresses more clearly. Thank you for your comment.

Reviewer #3 (Remarks to the Author):

The authors develop a new type of supramolecular elastomer material that can maintain excellent mechanical properties while possessing shape memory and self-healing functions, as well as clusterization-triggered emission effect. This tough multifunctional material is expected to be suitable for harsh working environments. In this manuscript, the authors introduce a new synthesis method that introduces metal coordination bonds. It retains excellent mechanical properties while possessing self-healing and shape memory properties. The authors conducted a large number of experiments as well as accurate analysis. In addition, the authors combined experiments with simulations, which added credibility to this work. However, several points need substantial improvement. My comments are:

Major issues:

1. In the manuscript, a large amount of testing and analysis were conducted on the mechanical properties of the materials. Whether the multifunctional properties of materials has been ignored? As the novelty of this work is the combination of mechanical properties and versatility multifunctional properties. There is a lack of principled explanation for self-healing and shape memory properties.

Response: We sincerely appreciate your constructive suggestions. According to your suggestions, we have supplemented and improved other properties of the material (shape memory, self-healing, clustering-triggered emission), for shape memory, we supplemented the shape memory demonstration of the material after shape reconstruction. For self-healing, we added self-healing tests of the materials at different conditions and explanations of relevant mechanisms. For clustering-triggered emission, we added an *in situ* fluorescence test. All of the above supplementary contents are shown below.

➤ Shape memory demonstration of the material after shape reconstruction.

Additionally, the original shape of PUU-0.5 could be remolded when the molecular chains were untangled and exchanged owing to the existence of dynamic

supramolecular interactions. **Fig. R16** (**Fig. S17** in Supplementary Information) presents the stress-relaxation curves of PUU-0.5 at different temperature. We observed that the rate of stress-relaxation is faster with the increase of temperature, so we chose 80 °C as the temperature for shape reconstruction because it effectively promotes bond exchange and topological rearrangement and complete shape reconstruction in less time. Subsequently, a rectangular and tiled PUU-0.5 specimen was reconstructed into a spring shape at 80 °C for 3 h, and the material was fully unfolded and maintained at −50 °C thereafter to freeze the temporary configuration (**Fig. R17**, **Fig. 5d** in revised manuscript). Subsequently, the reconstructed shape recovered to the spring shape rather than the initial rectangular configuration at 50 °C, indicating that PUU-0.5 exhibited shape-reconfigurable characteristics.

Fig. R16 (**Fig. S17** in Supplementary Information). The stress-relaxation curves of PUU-0.5 at different temperature.

Fig. R17 (Fig. 5d in revised manuscript). Shape-reconstructing properties of PUU-0.5.

➤ **Self-healing tests.**

Dynamic reversible coordination and hydrogen bonds endowed the synthesized materials with self-healing properties. Therefore, the self-healing properties of PUU-0.5—the specimen that exhibited optimal mechanical properties—were systematically examined. The simple thermally induced healing strategy was compared with another approach, which involved adding solvent drops to the fracture surface while it underwent heating; this was done to reducing the healing duration by accelerating the molecular chain motion. First, PUU-0 was healed at 40 °C for 2 h with and without N,N-dimethylformamide (DMF, a solvent used in synthetic materials). The solvent-free healed samples exhibited a poor strength (Fig. R18a, Fig. 6a in revised manuscript), resulting in a healing efficiency of only 30.31% (Eq. R1, Eq. 3 in revised manuscript). In contrast, the solvent-containing samples exhibited a higher healing efficiency (69.41%). Consequently, drops of DMF were added to a crack on PUU-0.5, and the sample was incubated at 40 °C for different durations. Analysis of the tensile strengths of the samples (Fig. R18b, Fig. 6b in revised manuscript) indicated that the material under the action of the solvent achieved a higher healing efficiency than that of the specimen healed using the simple thermal approach. Furthermore, the crack remained after the sample was healed at 40 °C for 2 h with DMF (Figs. R18 c and 18ci, Figs. 6c

and **6c₁** in revised manuscript). Nevertheless, the healing efficiency gradually increased with prolonged duration and reached 91.72% after 6 hours, following which the crack almost disappeared (**Figs. R18d** and **18d₁**, **Figs. 6d** and **6d₁** in revised manuscript). Owing to the presence of heat and the solvent, the molecular chain motion and the nondynamic covalent bond dissociation and recombination were accelerated (**Figs. R18 e** and **18e₁**, **Figs. 6e** and **6e₁** in revised manuscript), leading to crack repair. Analysis of the shape-memory properties of the healed PUU-0.5 (**Fig. R18f**, **Fig. 6f** in revised manuscript) indicated that the sample retained its original performance; moreover, the corresponding R_f (88.31%) and R_f (96.27%) values were almost identical to those of the original material. These results suggest that the healed PUU-0.5 retained its notable shape-memory performance.

Fig. R18 (Fig. 6 in revised manuscript) Assessment of self-healing behavior. (a) Tensile curves of pristine PUU-0, PUU-0 healed at 40 °C for 2 h, and PUU-0 healed at 40 °C for 2 h with DMF. (b) Tensile curves of pristine PUU-0.5 and those of PUU-0.5 healed at 40 °C for 2, 4, and 6 h with DMF. (c, c₁) Optical microscopic images of the fractured PUU-0.5 and PUU-0.5 healing at 40 °C for 2 h with DMF. (d, d₁) Optical microscopic images of the fractured PUU-0.5 and PUU-0.5 healing at 40 °C for 6 h with DMF. (e, e₁) Mechanism underlying the self-healing behavior of PUU-X. (f)

Shape-memory cycling curves of healed PUU-0.5.

The tensile strengths of the cracked and healed samples (σ_0 and σ , respectively) were used to calculate the healing efficiency (η) as follows:

$$\eta = \frac{\sigma}{\sigma_0} \times 100\%. \quad (\text{R1})$$

➤ **CTE Property of PUU-X.**

To validate the existence of this phenomenon, the fluorescence intensity of the material was recorded at different temperatures (Figs. R19 and R20, Figs. 8b and 8d in revised manuscript). The results indicated that the fluorescence intensity decreased gradually with increasing temperature; this was due to the luminescent groups no longer clustering at high temperatures owing to the dynamic bond dissociation. Therefore, the luminescence of the material was confirmed to be associated with dynamic bonds.

Fig. R19 (Fig. 8 b in revised manuscript) Photoluminescence spectra of PUU-0.5 at different temperatures.

Fig. R20 (Fig. 8d in revised manuscript) Photographs of PUU-0.5 acquired by on–off

switching a UV lamp at different temperatures.

2. An important aspect of this work is the addition of $\text{Zn}(\text{OTf})_2$. However, there is a lack of systematic discussion on its function and necessity in this manuscript. Two-dimensional infrared correlation spectrum is an interesting tool for analyzing dynamic bond changes. You have analyzed hydrogen bonds, can you also do the same for metal coordination bonds?

Response: Thanks for your careful review of the manuscript and your constructive comments. Since mechanical properties and self-healing ability of materials are mutually restricted, we need to consider many factors in the process of material design if we want to obtain better comprehensive properties of materials. Introducing non-dynamic covalent bonds into materials is an effective method to improve the mechanical properties of materials, among which hydrogen and coordination bonds are mostly studied. Because polyurethane contains partial hydrogen bonds, we only need a simple design for the hydrogen bonds, and how to design coordination bonds has become a major problem. Since the self-healing efficiency of polymer is determined by the kinetically labile of the cross-linking site, it is beneficial to design the kinetically labile of the cross-linking site to obtain the material with self-healing ability. It has been reported that the coordination of zinc ions with pyridine can form kinetically unstable cross-linking sites (*Adv. Mater.* 2020, 32, 1903762, *Macromol. Chem. Phys.* 2020, 221, 1900432), so in this work, we used the coordination of zinc ions with pyridine to obtain good self-healing properties.

For coordination bonds, UV-vis spectrophotometry was performed to corroborate the existence of coordination bonds in the synthesized materials. The absorption peaks of the Zn^{2+} -containing materials were red-shifted compared with those of PUU-0 (**Fig. R21, Fig. 2d** in revised manuscript) owing to the Zn^{2+} -pyridine coordination. The maximum absorption peak red-shifted progressively with increasing Zn^{2+} content (375 nm→397 nm→401 nm→405 nm→410 nm). Different degrees of Zn^{2+} -pyridine complexation were achieved with increasing Zn^{2+} content, which enhanced the electron ionization of the pyridine-ring-formed π - π conjugate system and reduced the energy required for transitioning, leading to a gradually increasing red shift⁴³.

Fig. R21 (Fig. 2d in revised manuscript) UV-vis absorption spectra of the PUU-X samples.

Detail issues:

1. It would be useful to include some comparison charts between this work and other works.

Response: Thanks for your valuable suggestion. According to your advice, we searched relevant works for comparison of mechanical properties, and drew a three-dimensional diagram, as shown in the **Fig. R22** (Fig. 2c in revised manuscript) below. In addition to this, we also compared the comprehensive properties of this material with other reported materials. As shown in the **Fig. R23** (Fig. 8c in revised manuscript), the material showed good comprehensive properties.

Fig. R22 (Fig. 2c in revised manuscript) Comparison of the strength, strain and toughness of PUU-0.5 with those reported works previously.

Fig. R23 (Fig.8c in revised manuscript) Comparison of this work with data from related reported work after normalization.

2. Annotations on illustrations can provide more information.

Response: Thank you for your constructive suggestion. We annotated some of the illustrations, as shown below.

Fig. R24 (Fig. 2d in revised manuscript) UV-vis absorption spectra of the PUU-X samples.

Fig. R25 (Fig. 2e in revised manuscript) Two-dimensional infrared (1755-1725 cm^{-1}) investigation of PUU-0.5.

Fig. R26 (Fig. 2f in revised manuscript) Two-dimensional infrared (1570-1530 cm^{-1}) investigation of PUU-0.5.

Fig. R27 (Fig. 7 in revised manuscript) **Shape-memory-assisted self-healing attributes of PUU-0.5.** (a) Side and (a₁) front views of PUU-0.5. (a₂) Schematic of self-healing at the fractured surfaces of the damaged sample. (b) Side and (b₁) front views of PUU-0.5 after healing for 2 h. (b₂) Illustration of progressive contact between the fractured surfaces of the damaged sample guided by the shape-memory effect. (c) Side and (c₁) front views of PUU-0.5 after healing for 4 h. (c₂) Schematic of the damaged sample being healed.

3. In Figure 3(as well as Figure 1), there should be two hydrogen bonds between two SDH molecules(D1), as is showed in D2.

Response: Thanks for your reminder and we apologize for this careless. We've corrected it in the revised manuscript, as shown below.

Fig. R28 Abridged general view and molecular structure of hydrogen and coordination bonds.

4. It is better to provide a locally enlarged image of Figure 2D and Figure 2E.

Response: Thanks for your kindly suggestion. We've enlarged Figure 2D (**Fig. R29**, **Fig. 2e** in revised manuscript) and Figure 2E (**Fig. R30**, **Fig. 2f** in revised manuscript), as shown below.

Fig. R29 (**Fig. 2e** in revised manuscript) Two-dimensional infrared (1755-1725 cm^{-1})

investigation of PUU-0.5.

Fig. R30 (Fig. 2f in revised manuscript) Two-dimensional infrared (1570-1530 cm^{-1}) investigation of PUU-0.5.

5. In table 1, “stress” and “strain” should be “Tensile strength” and “strain at break”.

Response: Thanks for your valuable suggestion. We’ve changed “Stress” and “Strain” to “Tensile strength” and “Strain at break”.

6. Line 97 and Line 101: It’s better to give the mole numbers of BIDI and DBTDL.

Response: Thank you for your review. We’ve added the mole numbers of BIDI and DBTDL in the revised manuscript on line 3 and 8, page 24.

7. Line 148: “ ϵ_B ” should be “ $\epsilon_{B,load}$ ”.

Response: Thanks for your careful review. We’ve changed “ ϵ_B ” to “ $\epsilon_{B,load}$ ”.

8. Line 193: “FTTR” should be “FTIR”. It is better to provide a locally enlarged image of Figure 2D and Figure 2E.

Response: We apologize for the error of spell. We’ve changed “FTTR” to “FTIR” and enlarged Figure 2D (**Fig. R29, Fig. 2e** in revised manuscript) and Figure 2E (**Fig. R30,**

Fig. 2f in revised manuscript).

9. Line 198: It is best not to use “poor stress”, as it still have good mechanical properties. It is better to use “worst tensile strength”.

Response: Thank you for your careful review of the manuscript. We’ve changed “poor stress” to “worst tensile strength”. After language polishing, we modified to “the most inferior tensile strength”.

Finally, we extremely appreciate the reviewer’s positive evaluation for our work. According to your suggestions and comments, the relevant descriptions have been revised, characterizations and contents have been supplemented. The main changes in the revised manuscript or supporting information are in red (revised contents) and blue (supplementary contents).

REVIEWERS' COMMENTS

Reviewer #1 (Remarks to the Author):

The authors have addressed my comments.

Reviewer #2 (Remarks to the Author):

The authors have sufficiently addressed the comments of the reviewers. I have no further comments.

Reviewer #3 (Remarks to the Author):

All concerns raised by the reviewers have been well addressed and thus the manuscript is now at a great state. Please consider to publish as is.